# Factorized visual representations in the primate visual system and deep neural networks

Jack W Lindsey[1,2], Elias B Issa[1,2]*

[1]Zuckerman Mind Brain Behavior Institute, Columbia University, New York, United States; [2]Department of Neuroscience, Columbia University, New York, United States

**Abstract** Object classification has been proposed as a principal objective of the primate ventral visual stream and has been used as an optimization target for deep neural network models (DNNs) of the visual system. However, visual brain areas represent many different types of information, and optimizing for classification of object identity alone does not constrain how other information may be encoded in visual representations. Information about different scene parameters may be discarded altogether ('invariance'), represented in non-interfering subspaces of population activity ('factorization') or encoded in an entangled fashion. In this work, we provide evidence that factorization is a normative principle of biological visual representations. In the monkey ventral visual hierarchy, we found that factorization of object pose and background information from object identity increased in higher-level regions and strongly contributed to improving object identity decoding performance. We then conducted a large-scale analysis of factorization of individual scene parameters – lighting, background, camera viewpoint, and object pose – in a diverse library of DNN models of the visual system. Models which best matched neural, fMRI, and behavioral data from both monkeys and humans across 12 datasets tended to be those which factorized scene parameters most strongly. Notably, invariance to these parameters was not as consistently associated with matches to neural and behavioral data, suggesting that maintaining non-class information in factorized activity subspaces is often preferred to dropping it altogether. Thus, we propose that factorization of visual scene information is a widely used strategy in brains and DNN models thereof.

*For correspondence:
elias.issa@columbia.edu

## eLife assessment

The study makes a **valuable** empirical contribution to our understanding of visual processing in primates and deep neural networks, with a specific focus on the concept of factorization. The analyses provide **convincing** evidence that high factorization scores are correlated with neural predictivity. This work will be of interest to systems neuroscientists studying vision and could inspire further research that ultimately may lead to better models of or a better understanding of the brain.

## Introduction

Artificial deep neural networks (DNNs) are the most predictive models of neural responses to images in the primate high-level visual cortex (*Cadieu et al., 2014*; *Schrimpf et al., 2020*). Many studies have reported that DNNs trained to perform image classification produce internal feature representations broadly similar to those in areas V4 and IT of the primate cortex, and that this similarity tends to be greater in models with better classification performance (*Yamins et al., 2014*). However, it remains opaque what aspects of the representations of these more performant models drive them to better match neural data. Moreover, beyond a certain threshold level of object classification performance,

**eLife digest** When looking at a picture, we can quickly identify a recognizable object, such as an apple, applying a single word label to it. Although extensive neuroscience research has focused on how human and monkey brains achieve this recognition, our understanding of how the brain and brain-like computer models interpret other complex aspects of a visual scene – such as object position and environmental context – remains incomplete.

In particular, it was not clear to what extent object recognition comes at the expense of other important scene details. For example, various aspects of the scene might be processed simultaneously. On the other hand, general object recognition may interfere with processing of such details.

To investigate this, Lindsey and Issa analyzed 12 monkey and human brain datasets, as well as numerous computer models, to explore how different aspects of a scene are encoded in neurons and how these aspects are represented by computational models. The analysis revealed that preventing effective separation and retention of information about object pose and environmental context worsened object identification in monkey cortex neurons. In addition, the computer models that were the most brain-like could independently preserve the other scene details without interfering with object identification.

The findings suggest that human and monkey high level ventral visual processing systems are capable of representing the environment in a more complex way than previously appreciated. In the future, studying more brain activity data could help to identify how rich the encoded information is and how it might support other functions like spatial navigation. This knowledge could help to build computational models that process the information in the same way, potentially improving their understanding of real-world scenes.

further improvement fails to produce a concomitant improvement in predicting primate neural responses (*Schrimpf et al., 2020*; *Nonaka et al., 2021*; *Linsley, 2023*). This weakening trend motivates finding new normative principles, besides object classification ability, that push models to better match primate visual representations.

One strategy for achieving high object classification performance is to form neural representations that discard some (are tolerant to) or all (are invariant to) information besides object class. Invariance in neural representations is in some sense a zero-sum strategy: building invariance to some parameters improves the ability to decode others. We also note that our use of 'invariance' in this context refers to invariance in neural representations, rather than behavioral or perceptual invariance (*DiCarlo and Cox, 2007*). However, high-level cortical neurons in the primate ventral visual stream are known to simultaneously encode many forms of information about visual input besides object identity, such as object pose (*Freiwald and Tsao, 2010*; *Hong et al., 2016*; *Kravitz et al., 2013*; *Peters and Kriegeskorte, 2021*). In this work, we seek to characterize how the brain simultaneously represents different forms of information.

In particular, we introduce methods to quantify the relationships between different types of visual information in a population code (e.g., object pose vs. camera viewpoint), and specifically the degree to which different forms of information are 'factorized'. Intuitively, if the variance driven by one parameter is encoded along orthogonal dimensions of population activity space compared to the variance driven by other scene parameters, we say that this representation is factorized. We note that our definition of factorization is closely related to the existing concept of manifold disentanglement (*DiCarlo and Cox, 2007*; *Chung et al., 2018*) and can be seen as a generalization of disentanglement to high-dimensional visual scene parameters like object pose. Factorization can enable simultaneous decoding of many parameters at once, supporting diverse visually guided behaviors (e.g., spatial navigation, object manipulation, or object classification) (*Johnston and Fusi, 2023*).

Using existing neural datasets, we found that both factorization of and invariance to object category and position information increase across the macaque ventral visual cortical hierarchy. Next, we leveraged the flexibility afforded by in silico models of visual representations to probe different forms of factorization and invariance in more detail, focusing on several scene parameters of interest: background content, lighting conditions, object pose, and camera viewpoint. Across a broad library of DNN models that varied in their architecture and training objectives, we found that factorization

of all of the above scene parameters in DNN feature representations was positively correlated with models' matches to neural and behavioral data. Interestingly, while neural invariance to some scene parameters (background scene and lighting conditions) predicted neural fits, invariance to others (object pose and camera viewpoint) did not. Our results generalized across both monkey and human datasets using different measures (neural spiking, fMRI, and behavior; 12 datasets total) and could not be accounted for by models' classification performance. Thus, we suggest that factorized encoding of multiple behaviorally relevant scene variables is an important consideration, alongside other desiderata such as classification performance, in building more brain-like models of vision.

## Results

Disentangling object identity manifolds in neural population responses can be achieved by qualitatively different strategies. These include building invariance of responses to non-identity scene parameters (or, more realistically, partial invariance; *DiCarlo and Cox, 2007*) and/or factorizing non-identity-driven response variance into isolated (factorized) subspaces (*Figure 1A*, left vs. center panels, cylindrical/spherical-shaded regions represent object manifolds). Both strategies maintain an 'identity subspace' in which object manifolds are linearly separable. In a non-invariant, non-factorized representation, other variables like camera viewpoint also drive variance within the identity subspace, 'entangling' the representations of the two variables (*Figure 1A*, right; viewpoint-driven variance is mainly in identity subspace, orange flat-shaded region).

To formalize these different representational strategies, we introduced measures of factorization and invariance to scene parameters in neural population responses (*Figure 1B*; see *Equations 2–4* in 'Methods'). Concretely, invariance to a scene variable (e.g., object motion) is computed by measuring the degree to which varying that parameter alone changes neural responses, relative to the changes induced by varying other parameters (lower relative influence on neural activity corresponds to higher invariance, or tolerance, to that parameter). Factorization is computed by identifying the axes in neural population activity space that are influenced by varying the parameter of interest and assessing how much it overlaps the axes influenced by other parameters ('*a*' in *Figure 1B and C*; lower overlap corresponds to higher factorization). We quantified this overlap in two different ways ('principal components analysis (PCA)-based' and 'covariance-based' factorization, corresponding to Equations 2 and 4 in 'Methods'), which produced similar results when compared in subsequent analyses (unless otherwise noted, factorization scores will generally refer to the PCA-based method, and the covariance method is shown in Figures 5–7 for comparison). Intuitively, a neural population in which one neural subpopulation encodes object identity and another separate subpopulation encodes object position exhibits a high degree of factorization of those two parameters (however, note that factorization may also be achieved by neural populations with mixed selectivity in which the 'subpopulations' correspond to subspaces, or independent orthogonal linear projections, of neural activity space rather than physical subpopulations). Though the example presented in *Figure 1* focused on factorization of and invariance to object identity versus non-identity variables, we stress that our definitions can be applied to any scene variables of interest. Furthermore, we presented a simplified visual depiction of the geometry within each scene variable subspace in *Figure 1*. We emphasize that our factorization metric does not require a particular geometry within a variable's subspace, whether parallel linearly ordered coding of viewpoint as in the cylindrical class manifolds shown in *Figure 1A and B*, or a more complex geometry where there is a lack of parallelism and/or a more nonlinear layout.

While factorization and invariance are not mutually exclusive representational strategies, they are qualitatively different. Factorization, unlike invariance, has the potential to enable the simultaneous representation of multiple scene parameters in a decodable fashion. Intuitively, factorization increases with higher dimensionality as this decreases overlap, all other things being equal (in the limit, the angle between points will approach 90° or a fully orthogonal code in high dimensions), and for a given finite, fixed dimension, factorization is mainly driven by the angle between this dimension and the other variable subspaces which measures the degree of contamination (*Figure 1C*; square vs. parallelogram). In a simulation, we found that the extent to which the variables of interest were represented in a factorized way (i.e., along orthogonal axes, rather than correlated axes) influenced the ability of a linear discriminator to successfully decode both variables in a generalizable fashion from a few training samples (*Figure 1C*).

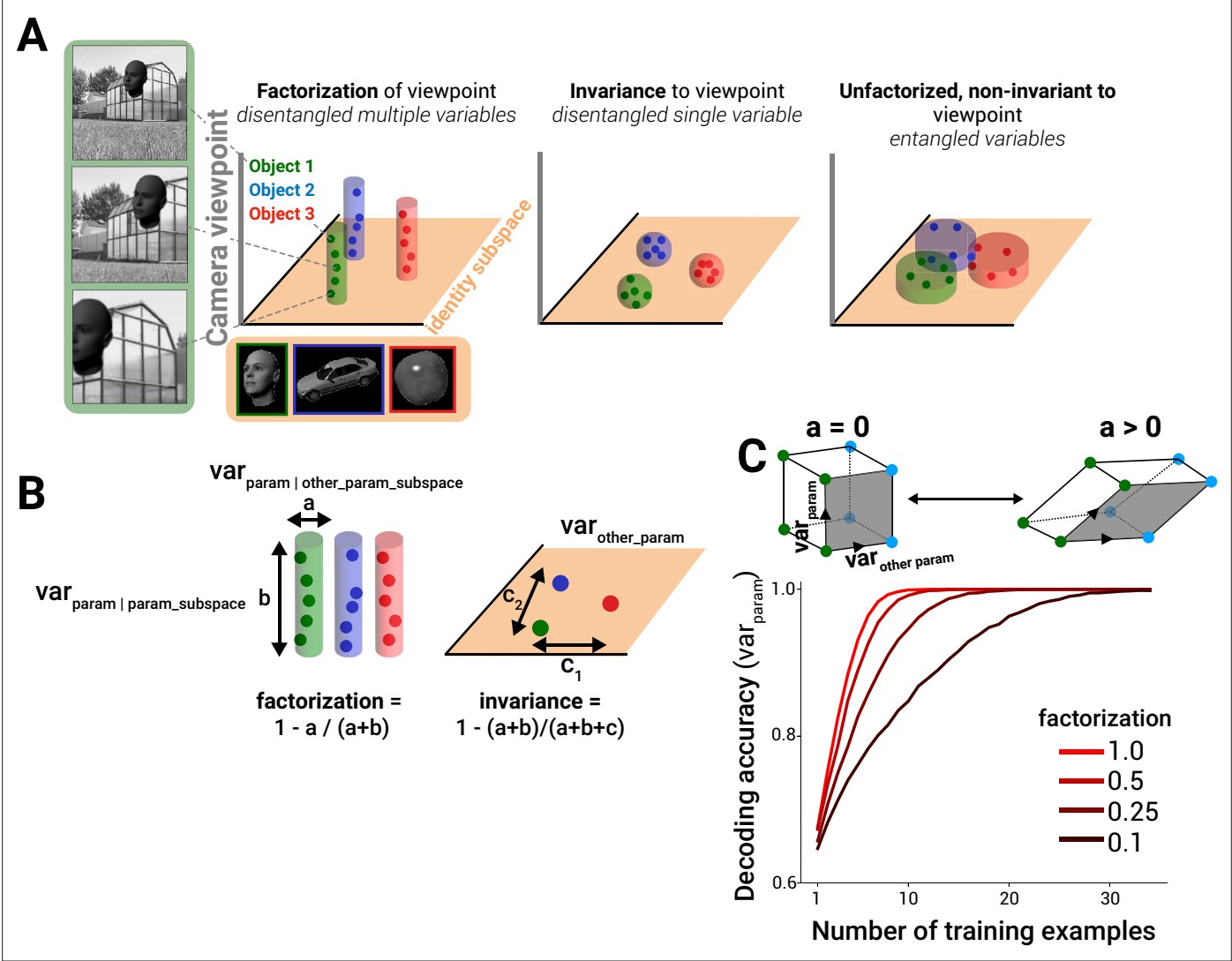

**Figure 1.** Framework for quantifying factorization in neural and model representations. (**A**) A subspace for encoding a variable, for example, object identity, in a linearly separable manner can be achieved by becoming invariant to non-class variables (compact spheres, middle column, where the volume of the sphere corresponds to the degree of neural invariance, or tolerance, for non-class variables; colored dots represent example images within each class) and/or by encoding variance induced by non-identity variables in orthogonal neural axes to the identity subspace (extended cylinders, left column). Only the factorization strategy simultaneously represents multiple variables in a disentangled fashion. A code that is sensitive to non-identity parameters within the identity subspace corrupts the ability to decode identity (right column) (identity subspace denoted by orange plane). (**B**) Variance across images within a class can be measured in two different linear subspaces: that containing the majority of variance for all other parameters ($a$, 'other_param_subspace') and that containing the majority of the variance for that parameter ($b$, 'param_subspace'). Factorization is defined as the fraction of parameter-induced variance that avoids the other-parameter subspace (left). By contrast, invariance to the parameter of interest is computed by comparing the overall parameter-induced variance to the variance in response to other parameters ($c$, 'var_other_param') (right). (**C**) In a simulation of coding strategies for two binary variables out of 10 total dimensions that are varying (see 'Methods'), a decrease in orthogonality of the relationship between the encoding of the two variables (alignment $a > 0$, or going from a square to a parallelogram geometry), despite maintaining linear separability of variables, results in poor classifier performance in the few training-samples regime when i.i.d. Gaussian noise is present in the data samples (only 3 of 10 dimensions used in simulation are shown).

Given the theoretically desirable properties of factorized representations, we next asked whether such representations are observed in neural data, and how much factorization contributes empirically to downstream decoding performance in real data. Specifically, we took advantage of an existing dataset in which the tested images independently varied object identity versus object pose plus background context (*Majaj et al., 2015*; https://github.com/brain-score/vision/blob/master/examples/ data_metrics_benchmarks.ipynb). We found that both V4 and IT responses exhibited more significant

factorization of object identity information from non-identity information than a shuffle control (which accounts for effects on factorization due to dimensionality of these regions) (*Figure 2—figure supplement 1*; see 'Methods'). Furthermore, the degree of factorization increased from V4 to IT (*Figure 2A*). Consistent with prior studies, we also found that invariance to non-identity information increased from V4 to IT in our analysis (*Figure 2A*, right, solid lines; *Rust and DiCarlo, 2010*). Invariance to non-identity information was even more pronounced when measured in the subspace of population activity capturing the bulk (90%) of identity-driven variance as a consequence of increased factorization of identity from non-identity information (*Figure 2A*, right, dashed lines).

To illustrate the beneficial effect of factorization on decoding performance, we performed a statistical lesion experiment that precisely targeted this aspect of representational geometry. Specifically, we analyzed a transformed neural representation obtained by rotating the population data so that inter-class variance more strongly overlapped with the principal components (PCs) of the intra-class variance in the data (see Equation 1 in 'Methods'). Note that this transformation, designed to decrease factorization, acts on the angle between latent variable subspaces. The applied linear basis rotation leaves all other activity statistics completely intact (such as mean neural firing rates, covariance structure of the population, and its invariance to non-class variables) yet has the effect of strongly reducing object identity decoding performance in both V4 and IT (*Figure 2B*). Our analysis shows that maintaining invariance alone in the neural population code was insufficient to account for a large fraction of decoding performance in high-level visual cortex; factorization of non-identity variables is key to the decoding performance achieved by V4 and IT representations.

We next asked whether factorization is found in DNN model representations and whether this novel, heretofore unconsidered metric, is a strong indicator of more brainlike models. When working with computational models, we have the liberty to test an arbitrary number of stimuli; therefore, we could independently vary multiple scene parameters at sufficient scale to enable computing factorization and invariance for each, and we explored factorization in DNN model representations in more depth than previously measured in existing neural experiments. To gain insight back into neural representations, we also assessed the ability of each model to predict separately collected neural and behavioral data. In this fashion, we may indirectly assess the relative significance of geometric properties like factorization and invariance to biological visual representations – if, for instance, models with more factorized representations consistently match neural data more closely, we may infer that those neural representations likely exhibit factorization themselves (*Figure 3*). To measure factorization, invariance, and decoding properties of DNN models, we generated an augmented image set, based on the images used in the previous dataset (*Figure 2*), in which we independently varied the foreground object identity, foreground object pose, background identity, scene lighting, and 2D scene viewpoint. Specifically for each base image from the original dataset, we generated sets of images that varied exactly one of the above scene parameters while keeping the others constant, allowing us to measure the variance induced by each parameter relative to the variance across all scene parameters (*Figure 3*, top left; 100 base scenes and 10 transformed images for each source of variation). We presented this large image dataset to models (4000 images total) to assess the relative degree of representational factorization of and invariance to each scene parameter. We conducted this analysis across a broad range of DNNs varying in architecture and objective as well as other implementational choices to obtain the widest possible range of DNN representations for testing our hypothesis. These included models using supervised training for object classification (*Krizhevsky et al., 2012*; *He et al., 2016*), contrastive self-supervised training (*He et al., 2020*; *Chen et al., 2020*), and self-supervised models trained using auxiliary objective functions (*Tian et al., 2019*; *Doersch et al., 2015*; *He et al., 2017*; *Donahue and Simonyan, 2019*; see 'Methods' and *Supplementary file 1b*).

First, we asked whether, in the course of training, DNN models develop factorized representations at all. We found that the final layers of trained networks exhibited consistent increases in factorization of all tested scene parameters relative to a randomly initialized (untrained) baseline with the same architecture (*Figure 4A*, top row, rightward shift relative to black cross, a randomly initialized ResNet-50). By contrast, training DNNs produced mixed effects on invariance, typically increasing it for background and lighting but reducing it for object pose and camera viewpoint (*Figure 4A*, bottom row, leftward shift relative to black cross for left two panels). Moreover, we found that the degree of factorization in models correlated with the degree to which they predicted neural activity for single-unit IT data (*Figure 4A*, top row), which can be seen as correlative evidence that neural representations in IT

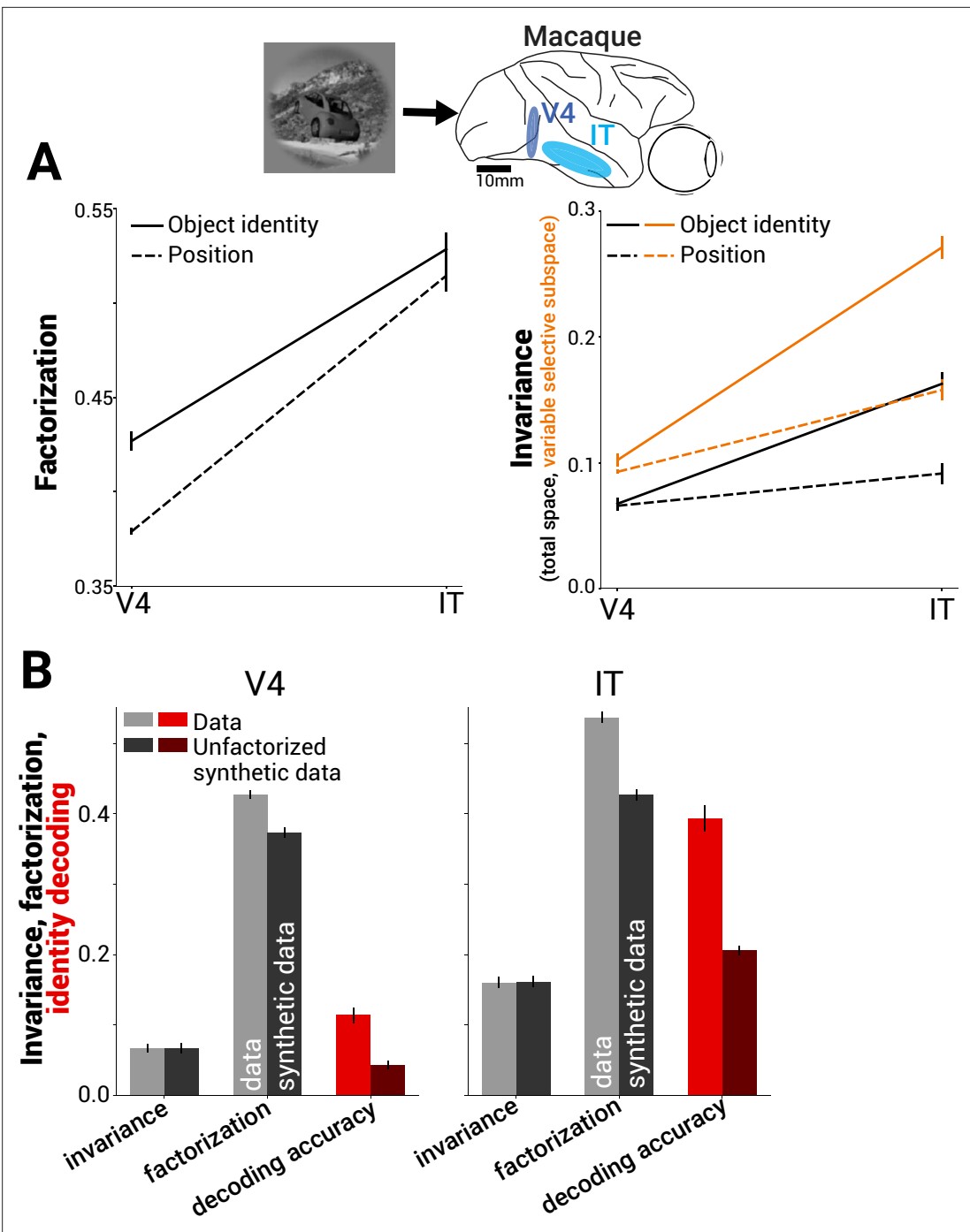

**Figure 2.** Benefit of factorization to neural decoding in macaque V4 and IT. (**A**) Factorization of object identity and position increased from macaque V4 to IT (PCA-based factorization, see 'Methods'; dataset E1 – multiunit activity in macaque visual cortex) (left). Like factorization, invariance also increased from V4 to IT (note, 'identity' refers to invariance to all non-identity position factors, solid black line) (right). Combined with increased factorization of the remaining variance, this led to higher invariance within the variable's subspace (orange lines), representing a neural subspace for identity information with invariance to nuisance parameters which decoders can target for read-out. (**B**) An experiment to test the importance of factorization for supporting object class decoding performance in neural responses. We applied a transformation to the neural data (linear basis rotation) that rotated the relative positions of mean responses to object classes without changing the relative proportion of within- vs. between-class variance (Equation 1 in 'Methods'). This transformation preserved invariance to non-class factors (leftmost pair of bars in each plot), while decreasing factorization of class information from non-class factors (center pair of bars in each plot). Concurrently, it had the effect of significantly reducing object class decoding performance (light vs. dark red bars in each plot, chance = 1/64; n = 128 multi-unit sites in V4 and 128 in IT).

The online version of this article includes the following figure supplement(s) for figure 2:

**Figure supplement 1.** Factorization and invariance in V4 and IT neural data.

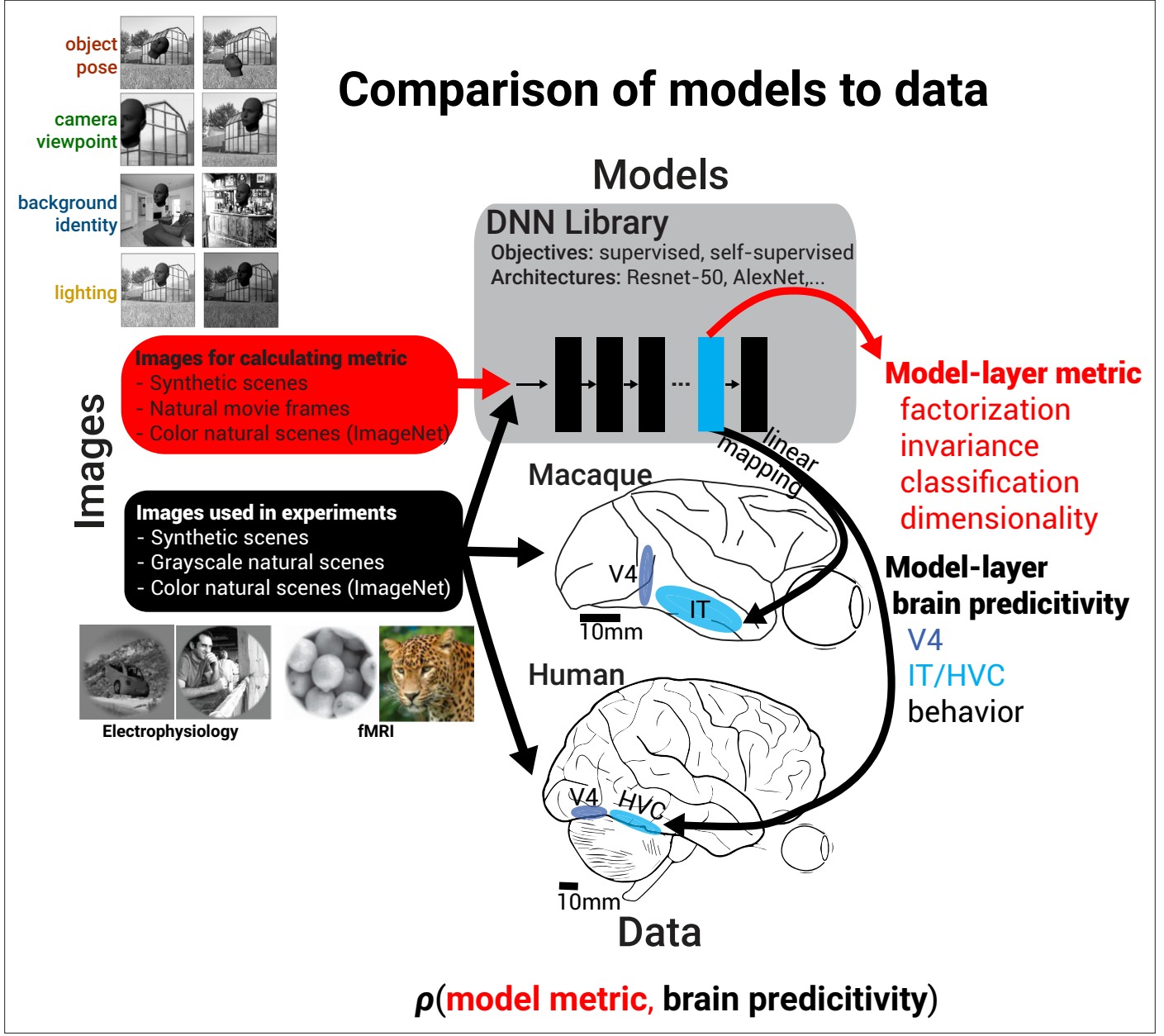

**Figure 3.** Measurement of factorization in deep neural network (DNN) models and comparison to brain data. Schematic showing how meta-analysis on models and brain data was conducted by first computing various representational metrics on models and then measuring a model's predictive power across a variety of datasets. For computing the representational metrics of factorization of and invariance to a scene parameter, variance in model responses was induced by individually varying each of four scene parameters (n = 10 parameter levels) for each base scene (n = 100 base scenes) (see images on the top left). The combination of model-layer metric and model-layer dataset predictivity for a choice of model, layer, metric, and dataset specifies the coordinates of a single dot on the scatter plots in *Figures 4 and 7*, and the across-model correlation coefficient between a particular representational metric and neural predictivity for a dataset summarizes the potential importance of the metric in producing more brainlike models (see *Figures 5 and 6*).

exhibit factorization of all scene variables tested. Interestingly, we saw a different pattern for representational invariance to a scene parameter. Invariance showed mixed correlations with neural predictivity (*Figure 4A*, bottom row), suggesting that IT neural representations build invariance to some scene information (background and lighting) but not to others (object pose and observer viewpoint). Similar effects were observed when we assessed correlations between these metrics and fits to human behavioral data rather than macaque neural data (*Figure 4B*).

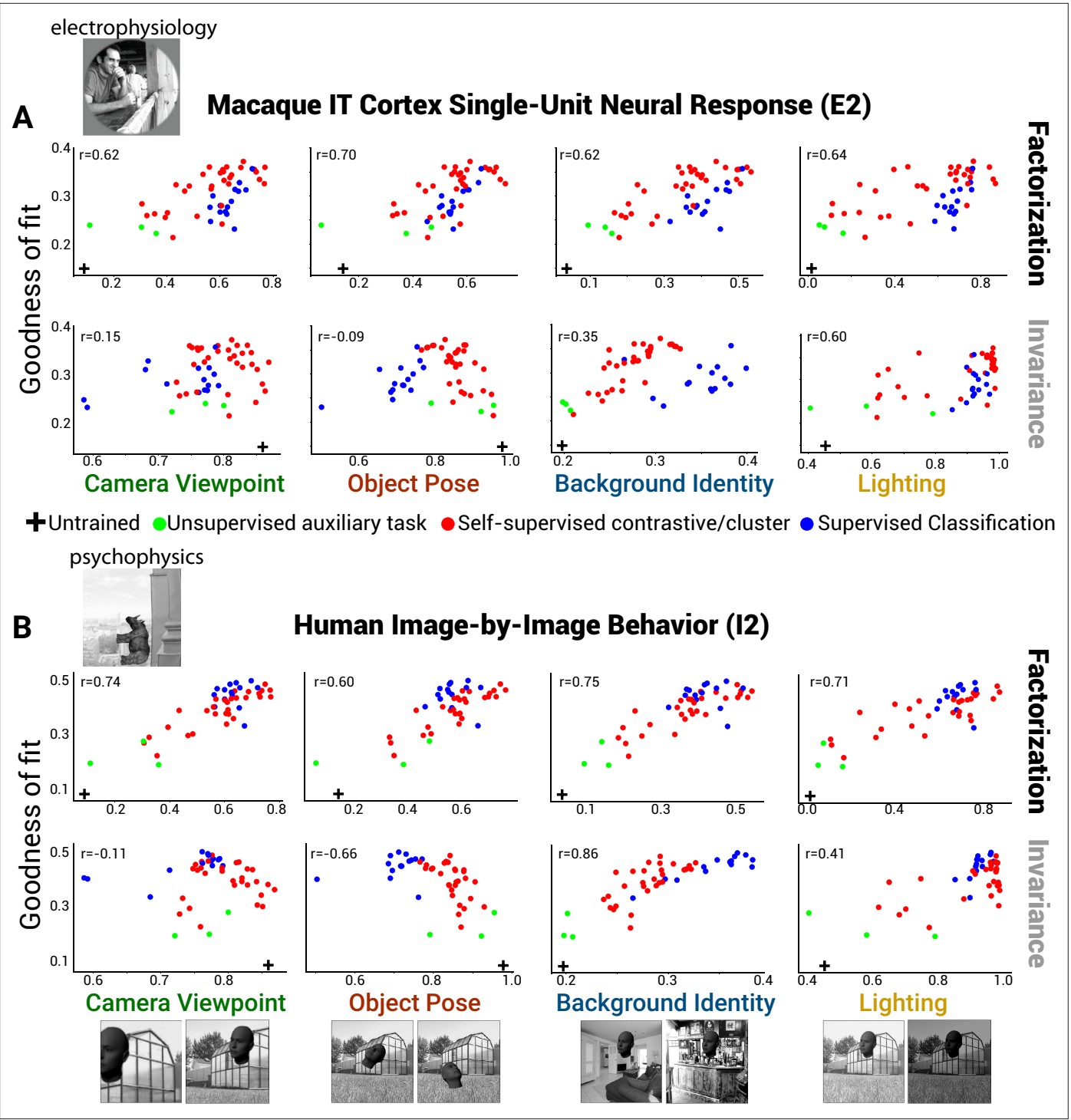

**Figure 4.** Neural and behavioral predictivity of models versus their factorization and invariance properties. (**A**) Scatter plots, for example, neural dataset (IT single units, macaque E2 dataset) showing the correlation between a model's predictive power as an encoding model for IT neural data versus a model's ability to factorize or become invariant to different scene parameters (each dot is a different model, using each model's penultimate layer). Note that factorization (PCA-based, see 'Methods') in trained models is consistently higher than that for an untrained, randomly initialized Resnet-50 DNN architecture (rightward shift relative to black cross). Invariance to background and lighting but not to object pose and viewpoint increased in trained models relative to the untrained control (rightward versus leftward shift relative to black cross). (**B**) Same as (**A**) except for human behavior performance patterns across images (human I2 dataset). Increasing scene parameter factorization in models generally correlated with better neural predictivity (top row). A noticeable drop in neural predictivity was seen for high levels of invariance to object pose (bottom row, second panel).

The online version of this article includes the following figure supplement(s) for figure 4:

*Figure 4 continued on next page*

*Figure 4 continued*

**Figure supplement 1.** Scatter plots for all datasets for V4.

**Figure supplement 2.** Scatter plots for all datasets for ITC/HVC.

**Figure supplement 3.** Scatter plots for all datasets for behavior.

To assess the robustness of these findings to choice of images and brain regions used in an experiment, we conducted the same analyses across a large and diverse set of previously collected neural and behavioral datasets, from different primate species and visual regions (six macaque datasets [*Majaj et al., 2015*; *Rust and DiCarlo, 2012*; *Rajalingham et al., 2018*]: two V4, two ITC (inferior temporal cortex), and two behavior; six human datasets [*Rajalingham et al., 2018*; *Kay et al., 2008*; *Shen et al., 2019*]: two V4, two HVC (higher visual cortex), and two behavior; *Supplementary file 1a*). Consistently, increased factorization of scene parameters in model representations correlated with models being more predictive of neural spiking responses, voxel BOLD signal, and behavioral responses to images (*Figure 5A*, black bars; see *Figure 4—figure supplements 1–3* for scatter plots across all datasets). Although invariance to appearance factors (background identity and scene lighting) correlated with more brainlike models, invariance for spatial transforms (object pose and camera viewpoint) consistently did not (zero or negative correlation values; *Figure 5C*, red and green open circles). Our results were preserved when we re-ran the analyses using only the subset of models with the identical ResNet-50 architecture (*Figure 5—figure supplement 1*) or when we evaluated model predictivity using representational dissimilarity matrices of the population (RDMs) instead of linear regression (encoding) fits of individual neurons or voxels (*Figure 5—figure supplement 2*). Furthermore, the main finding of a positive correlation between factorization and neural predictivity was robust to the particular choice of PCA threshold we used to quantify factorization (*Figure 5—figure supplement 3*). We found similar results using a covariance-based method for computing factorization that does not have any free parameters (*Figure 5C*, faded filled circles; see Equations 4 in 'Methods').

Finally, we tested whether our results generalized across the particular image set used for computing the model factorization scores in the first place. Here, instead of relying on our synthetically generated images, where each scene parameter was directly controlled, we re-computed factorization from two types of relatively unconstrained natural movies, one where the observer moves in an urban environment (approximates camera viewpoint changes) (*Lee et al., 2012*) and another where objects move in front of a fairly stationary observer (approximates object pose changes) (*Monfort, 2019*). Similar to the result found for factorization measured using augmentations of synthetic images, factorization of frame-by-frame variance (local in time, presumably dominated by either observer or camera motion; see 'Methods') from other sources of variance across natural movies (non-local in time) was correlated with improved neural predictivity in both macaque and human data while invariance to local frame-by-frame differences was not (*Figure 5B*; black versus gray bars). Thus, we have shown that a main finding – the importance of object pose and camera viewpoint factorization for achieving brainlike representations – holds across types of brain signal (spiking vs. BOLD), species (monkey vs. human), cortical brain areas (V4 vs. IT), images for testing in experiments (synthetic, grayscale vs. natural, color), and image sets for computing the metric (synthetic images vs. natural movies).

Our analysis of DNN models provides strong evidence that greater factorization of a variety of scene variables is consistently associated with a stronger match to neural and behavioral data. Prior work had identified a similar correlation between object classification performance (measured fitting a decoder for object class using model representations) and fidelity to neural data (*Yamins et al., 2014*). A priori, it is possible that the correlations we have demonstrated between scene parameter factorization and neural fit can be entirely captured by the known correlation between classification performance and neural fits (*Schrimpf et al., 2020*; *Yamins et al., 2014*) as factorization and classification may themselves be correlated. However, we found that factorization scores significantly boosted cross-validated predictive power of neural/behavioral fit performance compared to simply using object classification alone, and factorization boosted predictive power as much if not slightly more when using RDMs instead of linear regression fits to quantify the match to the brain/behavior (*Figure 6*). Thus, considering factorization in addition to object classification performance improves upon our prior understanding of the properties of more brainlike models (*Figure 7*).

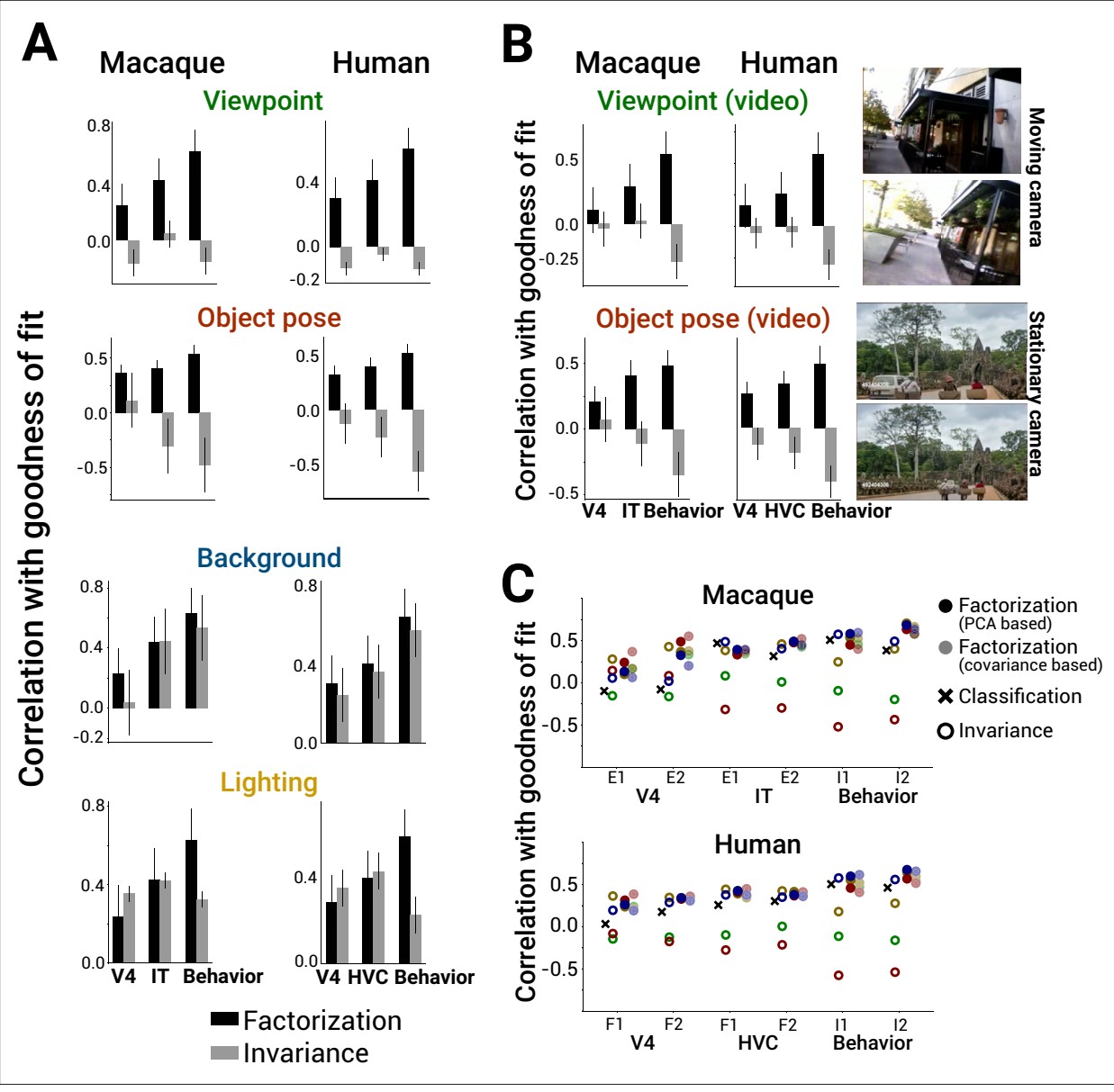

**Figure 5.** Scene parameter factorization correlates with more brainlike deep neural network (DNN) models. (**A**) Factorization of scene parameters in model representations computed using the PCA-based method consistently correlated with a model being more brainlike across multiple independent datasets measuring monkey neurons, human fMRI voxels, or behavioral performance in both macaques and humans (left vs. right column) (black bars). By contrast, increased invariance to camera viewpoint or object pose was not indicative of brainlike models (gray bars). In all cases, model representational metric and neural predictivity score were computed by averaging scores across the last 5 model layers. (**B**) Instead of computing factorization scores using our synthetic images (*Figure 3*, top left), recomputing camera viewpoint or object pose factorization from natural movie datasets that primarily contained camera or object motion, respectively, gave similar results for predicting which model representations would be more brainlike (right: example movie frames; also see 'Methods'). Error bars in (**A and B**) are standard deviations over bootstrapped resampling of the models. (**C**) Summary of the results from (**A**) across datasets (x-axis) for invariance (open symbols) versus factorization (closed symbols) (for reference, '*x*' symbols indicate predictive power when using model classification performance). Results using a comparable, alternative method for computing factorization (covariance-based, Equation 4 in 'Methods'; light closed symbols) are shown adjacent to the original factorization metric (PCA-based, Equation 2 in 'Methods'; dark closed symbols).

The online version of this article includes the following figure supplement(s) for figure 5:

**Figure supplement 1.** Predictivity of factorization and invariance restricting to ResNet-50 model architectures.

**Figure supplement 2.** Predictivity of factorization and invariance for representational dissimilarity matrices (RDMs).

**Figure supplement 3.** Effect on neural and behavioral predictivity of PCA threshold for computing PCA-based factorization, related to *Figure 5*.

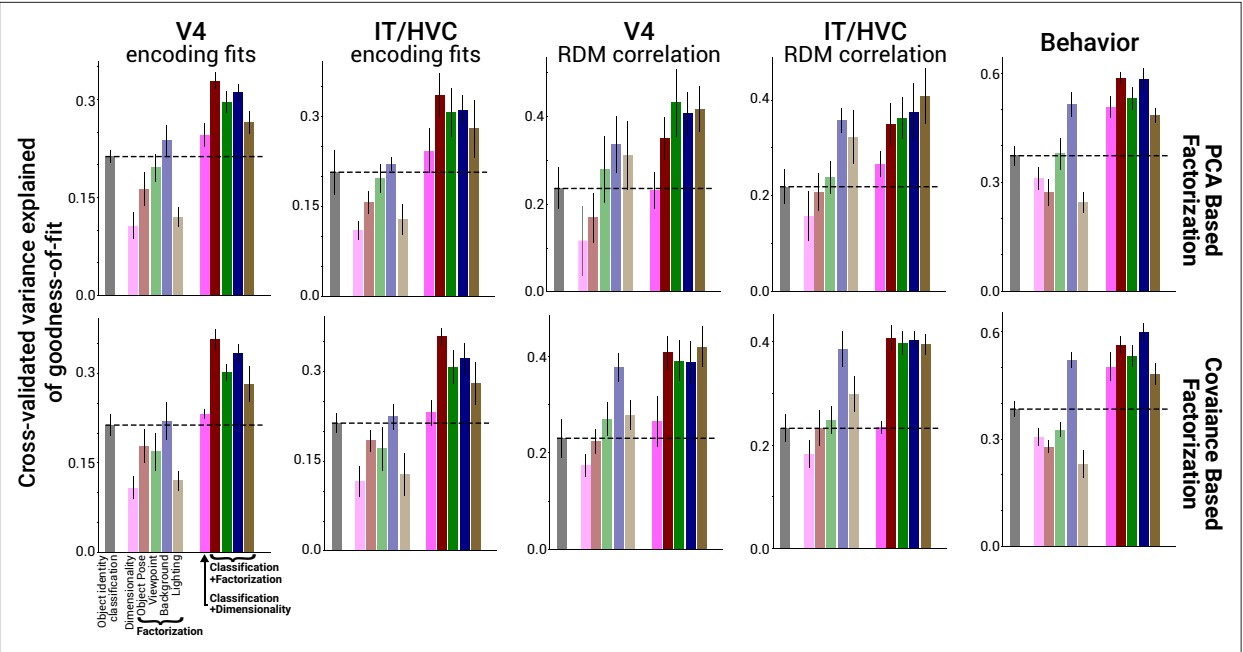

**Figure 6.** Scene parameter factorization combined with object identity classification improves correlations with neural predictivity. Average across datasets of brain predictivity of classification (faded black bar), dimensionality (faded pink bar), and factorization (remaining faded colored bars) in a model representation. Linearly combining factorization with classification in a regression model (unfaded bars at right) produced significant improvements in predicting the most brainlike models (performance cross-validated across models and averaged across datasets, n = 4 datasets for each of V4, IT/HVC and behavior). The boost from factorization in predicting the most brainlike models was not observed for neural and fMRI data when combining classification with a model's overall dimensionality (solid pink bars; compared to black dashed line for brain predictivity when using classification alone). Results are shown for both the PCA-based and covariance-based factorization metric (top versus bottom row). Error bars are standard deviations over bootstrapped resampling of the models.

## Discussion

Object classification, which has been proposed as a normative principle for the function of the ventral visual stream, can be supported by qualitatively different representational geometries (*Yamins et al., 2014*; *Nayebi, 2021*). These include representations that are completely invariant to non-class information (*Caron et al., 2019b*; *Caron, 2019a*) and representations that retain a high-dimensional but factorized encoding of non-class information, which disentangles the representation of multiple variables (*Figure 1A*). Here, we presented evidence that factorization of non-class information is an important strategy used, alongside invariance, by the high-level visual cortex (*Figure 2*) and by DNNs that are predictive of primate neural and behavioral data (*Figures 4 and 5*).

Prior work has indicated that building representations that support object classification performance and representations that preserve high-dimensional information about natural images are both important principles of the primate visual system (*Cadieu et al., 2014*; *Elmoznino and Bonner, 2022*; though see *Conwell et al., 2022*). Critically, our results cannot be accounted for by classification performance or dimensionality alone (*Figure 6*, gray and pink bars); that is, the relationship between factorization and matches to neural data was not entirely mediated by classification or dimensionality. That said, we do not regard factorization and dimensionality, or factorization and object classification performance, as mutually exclusive hypotheses for useful principles of visual representations. Indeed, high-dimensional representations could be regarded as a means to facilitate factorization, and likewise factorized representations can better support classification (*Figure 1C*).

Our notion of factorization is related to, but distinct from, several other concepts in the literature. Many prior studies in machine learning have considered the notion of disentanglement, often defined as the problem of inferring independent factors responsible for generating the observed data (*Kim and Mnih, 2018*; *Eastwood and Williams, 2018*; *Higgins, 2018*). One prior study notably found that machine learning models designed to infer disentangled representations of visual data displayed single-unit responses that resembled those of individual neurons in macaque IT (*Higgins et al., 2021*).

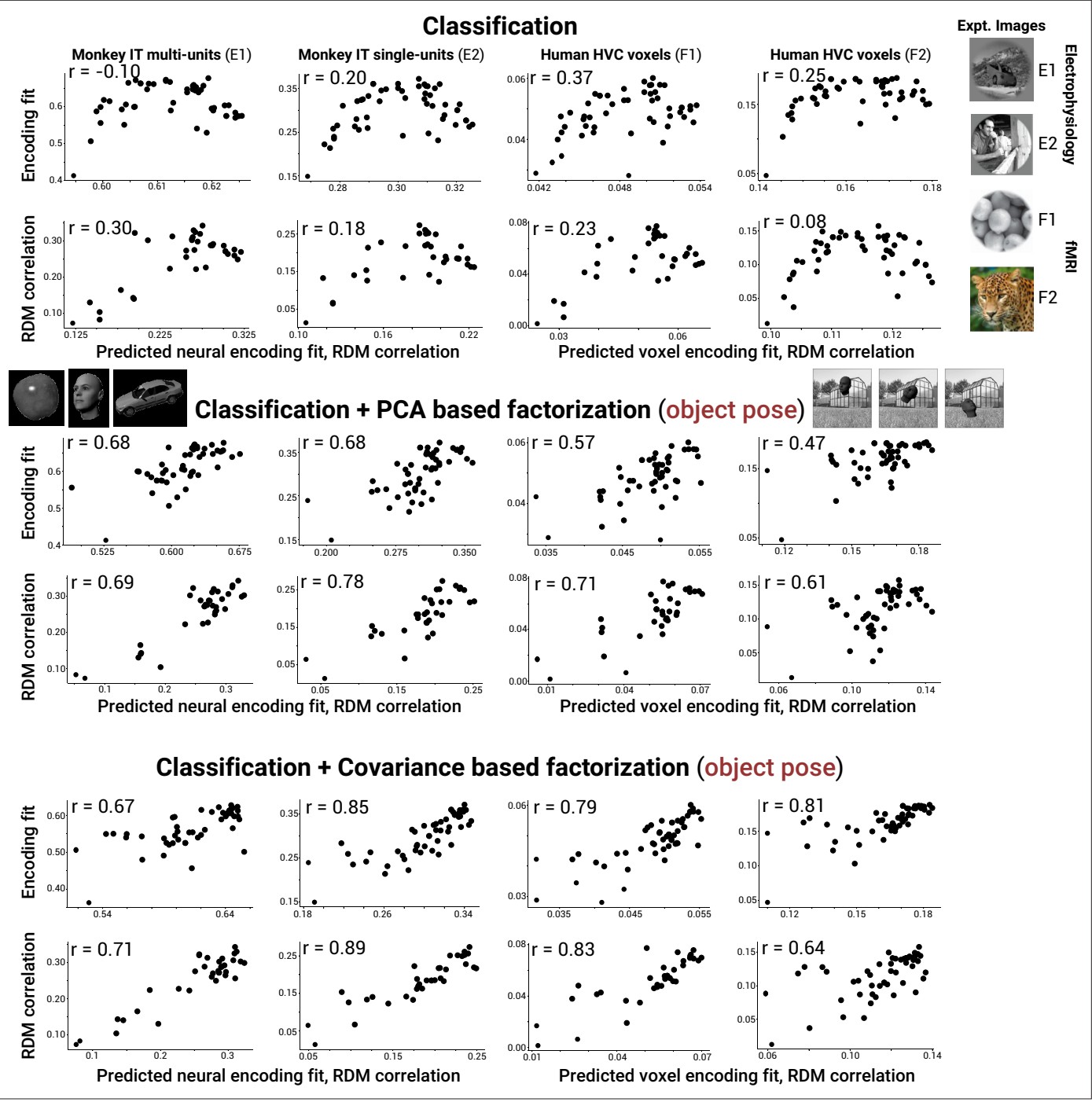

**Figure 7.** Combining classification performance with object pose factorization improves predictions of the most brainlike models on IT/HVC data. Example scatter plots for neural and fMRI datasets (macaque E1 and E2, IT multi units and single units; human F1 and F2, fMRI voxels) showing a saturating and sometimes reversing trend in neural (voxel) predictivity for models that are increasingly good at classification (top row). This saturating/reversing trend is no longer present when adding object pose factorization to classification as a combined, predictive metric for brainlikeness of a model (middle and bottom rows). The x-axis of each plot indicates the predicted encoding fit or representational dissimilarity matrix (RDM) correlation after fitting a linear regression model with the indicated metrics as input (either classification or classification + factorization).

Our definition of factorization is more flexible, requiring only that independent factors be encoded in orthogonal subspaces, rather than by distinct individual neurons. Moreover, our definition applies to generative factors, such as camera viewpoint or object pose, that are multidimensional and context dependent. Factorization is also related to a measure of 'abstraction' in representational geometry

introduced in a recent line of work (*Bernardi et al., 2020*; *Boyle et al., 2024*), which is observed to emerge in trained neural networks (*Johnston and Fusi, 2023*; *Alleman et al., 2024*). In these studies, an abstract representation is defined as one in which variables are encoded and can be decoded in a consistent fashion regardless of the values of other variables. A fully factorized representation should be highly abstract according to this definition, though factorization emphasizes the geometric properties of the population representation while these studies emphasize the consequences for decoding performance in training downstream linear read-outs. Relatedly, another recent study found that orthogonal encoding of class and non-class information is one of several factors that determines few-shot classification performance (*Sorscher et al., 2022*). Our work can be seen as complementary to work on representational straightening of natural movie trajectories in the population space (*Hénaff et al., 2021*). This work suggested that visual representations maintain a locally linear code of latent variables like camera viewpoint, while our work focused on the global arrangement of the linear subspaces affected by different variables (e.g., overall coding of camera viewpoint-driven variance versus sources of variance from other scene variables in a movie). Local straightening of natural movies was found to be important for early visual cortex neural responses but not necessarily for high-level visual cortex (*Toosi and Issa, 2022*), where the present work suggests factorization may play a role.

Our work has several limitations. First, our analysis is primarily correlative. Going forward, we suggest that factorization could prove to be a useful objective function for optimizing neural network models that better resemble primate visual systems, or that factorization of latent variables should at least be a by-product of other objectives that lead to more brain-like models. An important direction for future work is finding ways to directly incentivize factorization in model objective functions so as to test its causal impact on the fidelity of learned representations to neural data. Second, our choice of scene variables to analyze in this study was heuristic and somewhat arbitrary. Future work could consider unsupervised methods (in the vein of independent components analysis) for uncovering the latent sources of variance that generate visual data, and assessing to what extent these latent factors are encoded in factorized form. Third, in our work we do not specify the details of how a particular scene parameter is encoded within its factorized subspace, including whether the code is linear ('straightened') or nonlinear (*Hénaff et al., 2021*; *Hénaff et al., 2019*). Neural codes could adopt different strategies, resulting in similar factorization scores at the population level, each with some support in visual cortex literature: (1) each neuron encodes a single latent variable (*Field, 1994*; *Chang and Tsao, 2017*), (2) separate brain subregions encode qualitatively different latent variables but using distributed representations within each region (*Tsao et al., 2006*; *Lafer-Sousa and Conway, 2013*; *Vaziri et al., 2014*), and (3) each neuron encodes multiple variables in a distributed population code, such that the factorization of different variables is only apparent as independent directions when assessed in high-dimensional population activity space (*Field, 1994*; *Rigotti et al., 2013*). Future work can disambiguate among these possibilities by systematically examining ventral visual stream subregions (*Kravitz et al., 2013*; *Vaziri et al., 2014*; *Kravitz et al., 2011*) and the single neuron tuning curves within them (*Leopold et al., 2006*; *Freiwald et al., 2009*).

## Methods
### Monkey datasets
Macaque monkey datasets were of single-unit neural recordings (*Rust and DiCarlo, 2012*), multi-unit neural recordings (*Majaj et al., 2015*), and object recognition behavior (*Rajalingham et al., 2018*). Single-unit spiking responses to natural images were measured in V4 and anterior ventral IT (*Rust and DiCarlo, 2012*). The advantages of this dataset are that it contains well-isolated single neurons, the gold standard for electrophysiology. Furthermore, the IT recordings were obtained from penetrating electrodes targeting the anterior ventral portion of IT near the base of skull, reflecting the highest level of the IT hierarchy. On the other hand, the multi-unit dataset was obtained from across IT with a bias toward where multi-unit arrays are more easily placed such as CIT and PIT (*Majaj et al., 2015*), complementing the recording locations of the single-unit dataset. An advantage of the multi-unit dataset using chronic recording arrays is that an order of magnitude more images were tested per recording site (see dataset comparisons in *Supplementary file 1a*). Finally, the monkey behavioral dataset came from a third study examining the image-by-image object classification performance of macaques and humans (*Rajalingham et al., 2018*).

## Human datasets

Three datasets from humans were used, two fMRI datasets and one object recognition behavior dataset (*Nonaka et al., 2021*; *Rajalingham et al., 2018*; *Kay et al., 2008*). The fMRI datasets used different images (color versus grayscale) but otherwise used a fairly similar number of images and voxel resolution in MR imaging. Human fMRI studies have found that different DNN layers tend to map to V4 and HVC human fMRI voxels (*Nonaka et al., 2021*). The human behavioral dataset measured image-by-image classification performance and was collected in the same study as the monkey behavioral signatures (*Rajalingham et al., 2018*).

## Computational models

In recent years, a variety of approaches to training DNN vision models have been developed that learn representations that can be used for downstream classification (and other) tasks. Models differ in a variety of implementational choices including in their architecture, objective function, and training dataset. In the models we sampled, objectives included supervised learning of object classification (AlexNet, ResNet), self-supervised contrastive learning (MoCo, SimCLR), and other unsupervised learning algorithms based on auxiliary tasks (e.g., reconstruction or colorization). A majority of the models that we considered relied on the widely used, performant ResNet-50 architecture, though some in our library utilized different architectures. The randomly initialized network control utilized ResNet-50 (see *Figure 4A and B*). The set of models we used is listed in *Supplementary file 1b*.

## Simulation of factorized versus non-factorized representational geometries

For the simulation in *Figure 1C*, we generated data in the following way. First, we randomly sampled the values of N = 10 binary features. Feature values corresponded to positions in an N-dimensional vector space as follows: each feature was assigned an axis in N-dimensional space, and the value of each feature (+1 or –1) was treated as a coefficient indicating the position along that axis. All but two of the feature axes were orthogonal to the rest. The last two features, which served as targets for the trained linear decoders, were assigned axes whose alignment ranged from 0 (orthogonal) to 1 (identical). In the noiseless case, factorization of these two variables with respect to one another is given by subtracting the square of the cosine of the angle between the axes from 1. We added Gaussian noise to the positions of each data point and randomly sampled K positive and negative examples for each variable of interest to use as training data for the linear classifier (a support vector machine).

## Macaque neural data analyses

For the shuffle control used as a null model for factorization, we shuffled the object identity labels of the images (*Figure 2—figure supplement 1*). For the transformation used in *Figure 2B*, we computed the PCs of the mean neural activity response to each object class ('class centers,' $x^c$), referred to as the inter-class PCs, $v_1^{inter}$, $v_2^{inter}$, …, $v_N^{inter}$. We also computed the PCs of the data with corresponding class centers subtracted (i.e., $x - x^c$) from each activity pattern, referred to as the intra-class PCs $v_1^{intra}$, $v_2^{intra}$, …, $v_N^{intra}$. We transformed the data by applying to the class centers a change of basis matrix $W_{inter \to intra}$ that rotated each inter-class PC into the corresponding intra-class PC: $W_{inter \to intra} = v_1^{intra} (v_1^{inter})^T$ + …$1 v_N^{intra} (v_N^{inter})^T$. That is, the class centers were transformed by this matrix, but the relative positions of activity patterns within a given class were fixed. For an activation vector $x$ belonging to a class c for which the average activity vector over all images of class c is $x^c$, the transformed vector was

$$x_{\text{transformed}} = W_{inter \to intra} \, x^c + (x - x^c) \tag{1}$$

This transformation has the effect of preserving intra-class variance statistics exactly from the original data and preserving everything about the statistics of inter-class variance except its orientation relative to intra-class variance. That is, the transformation is designed to affect (specifically decrease) factorization while controlling for all other statistics of the activity data that may be relevant to object classification performance (considering the simulation in *Figure 1C* of two binary variables, this basis change of the neural data in *Figure 2B* is equivalent to turning a square into the maximally flat parallelogram, the degenerate one where all the points are collinear).

## Scene parameter variation

Our generated scenes consisted of foreground objects imposed upon natural backgrounds. To measure variance associated with a particular parameter like the background identity, we randomly sampled 10 different backgrounds while holding the other variables (e.g., foreground object identity and pose constant). To measure variance associated with foreground object pose, we randomly varied object angle from [–90, 90] along all three axes independently, object position on the two in-plane axes, horizontal [–30%, 30%] and vertical [–60%, 60%], and object size [×1/1.6, ×1.6]. To measure variance associated with camera position, we took crops of the image with scale uniformly varying from 20 to 100% of the image size, and position uniformly distributed across the image. To measure variance associated with lighting conditions, we applied random jitters to the brightness, contrast, saturation, and hue of an image, with jitter value bounds of [–0.4, 0.4] for brightness, contrast, and saturation and [–0.1, 0.1] for hue. These parameter choices follow standard data augmentation practices for self-supervised neural network training, as used, for example, in the SimCLR and MoCo models tested here (*He et al., 2020*; *Chen et al., 2020*).

## Factorization and invariance metrics

Factorization and invariance were measured according to the following equations:

$$factorization_{param} = 1 - \text{var}_{param|other\_param\_subspace}/\text{var}_{param} \tag{2}$$

$$invariance_{param} = 1 - \text{var}_{param}/\text{var}_{all\ param} \tag{3}$$

Variance induced by a parameter ($var_{param}$) is computed by measuring the variance (summed across all dimensions of neural activity space) of neural responses to the 10 augmented versions of a base image where the augmentations are those obtained by varying the parameter of interest. This quantity is then averaged across the 100 base images. The variance induced by all parameters is simply the sum of the variances across all images and augmentations. To define the 'other-parameter subspace,' we averaged neural responses for a given base image over all augmentations using the parameter of interest, and ran PCA on the resulting set of averaged responses. The subspace was defined as the space spanned by top PCA components containing 90% of the variance of these responses. Intuitively, this space captures the bulk of the variance driven by all parameters other than the parameter of interest (due to the averaging step). The variance of the parameter of interest *within* this 'other-parameter subspace,' $var_{param|other\_param\_subspace}$, was computed the same way as $var_{param}$ but using the projections of neural activity responses onto the other-parameter subspace. In the main text, we refer to this method of computing factorization as PCA-based factorization.

We also considered an alternative definition of factorization referred to as covariance-based factorization. In this alternative definition, we measured the covariance matrices $cov_{param}$ and $cov_{other\_param}$ induced by varying (in the same fashion as above) the parameter of interest, and all other parameters. Factorization was measured by the following equation:

$$factorization_{param} = 1 - \text{Trace}[(\text{cov}_{param})^{\text{T}} \text{cov}_{other\_param}]/(\text{Trace}[(\text{cov}_{param})^{\text{T}} \text{cov}_{param}]$$
$$\text{Trace}[(\text{cov}_{other\_param})^{\text{T}} \text{cov}_{other\_param}])^{1/2} \tag{4}$$

This is equal to 1 minus the dot product between the normalized, flattened covariance matrices, and thus covariance-based factorization is a measure of the discrepancy of the covariance structure induced by the parameter of interest and other parameters. The main findings were unaffected by our choice of method for computing the factorization metric, whether PCA or covariance based (*Figures 5–7*). An advantage of the PCA-based method is that as an intermediate one recovers the linear subspaces containing parameter variance, but in so doing requires an arbitrary choice of the explained variance threshold used to choose the number of PCs. By contrast, the covariance-based method is more straightforward to compute and has no free parameters. Thus, these two metrics are complementary and somewhat analogous in methodology to two metrics commonly used for measuring dimensionality (the number of components needed to explain a certain fraction of the

variance, analogous to our original PCA-based definition, and the participation ratio, analogous to our covariance-based definition) (*Ding and Glanzman, 2010*; *Litwin-Kumar et al., 2017*).

## Natural movie factorization metrics

For natural movies, variance is not induced by explicit control of a parameter as in our synthetic scenes but implicitly, by considering contiguous frames (separated by 200 ms in real time) as reflective of changes in one of two motion parameters (object versus observer motion) depending on how stationary the observer is (MIT Moments in Time movie set: stationary observer; UT-Austin Egocentric movie set: nonstationary) (*Lee et al., 2012*; *Monfort, 2019*). Here, the *all parameters* condition is simply the variance across all movie frames, which in the case of MIT Moments in Time dataset includes variance across thousands of video clips taken in many different settings and in the case of the UT-Austin Egocentric movie dataset includes variance across only four movies but over long durations of time during which an observer translates extensively in an environment (3–5 hr). Thus, movie clips in the MIT Moments in Time movie set contained new scenes with different object identities, backgrounds, and lightings and thus effectively captured variance induced by these non-spatial parameters (*Monfort, 2019*). In the UT-Austin Egocentric movie set, new objects and backgrounds are encountered as the subject navigates around the urban landscape (*Lee et al., 2012*).

## Model neural encoding fits

Linear mappings between model features and neuron (or voxel) responses were computed using ridge regression (with regularization coefficient selected by cross-validation) on a low-dimensional linear projection of model features (top 300 PCA components computed using images in each dataset). We also tested an alternative approach to measuring representational similarity between models and experimental data based on representational similarity analysis (*Kriegeskorte and Kievit, 2013*), computing dot product similarities of the representations of all pairs of images and measuring the Spearman correlation coefficient between these pairwise similarity matrices obtained from a given model and neural dataset, respectively.

## Model behavioral signatures

We followed the approach of *Rajalingham et al., 2018*. We took human and macaque behavioral data from the object classification task and used it to create signatures of image-level difficulty (the 'I1' vector) and image-by-distractor-object confusion rates (the 'I2' matrix). We did the same for the DNN models, extracting model 'behavior' by training logistic regression classifiers to classify object identity in the same image dataset used in the experiments of *Rajalingham et al., 2018*, using model layer activations as inputs. Model behavioral accuracy rates on image by distractor object pairs were assessed using the classification probabilities output by the logistic regression model, and these were used to compute I1 and I2 metrics as was done for the true behavioral data. Behavioral similarity between models and data was assessed by measuring the correlation between the entries of the I1 vectors and I2 matrices (both I1 and I2 results are reported).

## Model layer choices

The scatter plots in *Figure 4A and B* and *Figure 4—figure supplements 1–3* use metrics (factorization, invariance, and goodness of neural fit) taken from the final representational layer of the network (the layer prior to the logits layer used for classification in supervised network, prior to the embedding head in contrastive learning models, or prior to any auxiliary task-specific layers in unsupervised models trained using auxiliary tasks). However, representational geometries of model activations, and their match to neural activity and behavior, vary across layers. This variability arises because different model layers correspond to different stages of processing in the model (convolutional layers in some cases, and pooling operations in others), and may even have different dimensionalities. To ensure that our results do not depend on idiosyncrasies of representations in one particular model layer and the particular network operations that precede it, summary correlation statistics in all other figures (*Figures 5–7*, *Figure 5—figure supplements 1–3*) show the results of the analysis in question averaged over the five final representational layers of the model. That is, the metrics of interest (factorization, invariance, neural encoding fits, RDM correlation, behavioral similarity scores) were computed

independently for each of the five final representational layers of each model, and these five values were averaged prior to computing correlations between different metrics.

## Correlation of model predictions and experimental data

A Spearman linear correlation coefficient was calculated for each model layer by biological dataset combination (six monkey datasets and six human datasets). Here, we do not correct for noise in the biological data when computing the correlation coefficient as this would require trial repeats (for computing intertrial variability) that were limited or not available in the fMRI data used. In any event, normalizing by the data noise ceiling applies a uniform scaling to all model prediction scores and does not affect model comparison, which only depends on ranking models as being relatively better or worse in predicting brain data. Finally, we estimated the effectiveness of model factorization, invariance, or dimensionality in combination with model object classification performance for predicting model neural and behavioral fit by performing a linear regression on the particular dual metric combination (e.g., classification plus object pose factorization) and reporting the Spearman correlation coefficient of the linearly weighted metric combination. The correlation was assessed on held-out models (80% used for training, 20% for testing), and the results were averaged over 100 randomly sampled train/test splits.

## Acknowledgements

This work was performed on the Columbia Zuckerman Institute Axon GPU cluster and via generous access to Cloud TPUs from Google's TPU Research Cloud (TRC). JWL was supported by the DOE CSGF (DE-SC0020347). EBI was supported by a Klingenstein-Simons fellowship, Sloan Foundation fellowship, and Grossman-Kavli Scholar Award. We thank Erica Shook for comments on a previous version of the manuscript. The authors declare no competing interests.

## Additional information

### Funding

| Funder | Grant reference number | Author |
|---|---|---|
| DOE CSGF | DE-SC0020347 | Jack W Lindsey |
| Klingenstein-Simons Foundation | Fellowship in Neuroscience | Elias B Issa |
| Sloan Foundation | Fellowship | Elias B Issa |
| Grossman-Kavli Center at Columbia | Scholar Award | Elias B Issa |

The funders had no role in study design, data collection and interpretation, or the decision to submit the work for publication.

### Author contributions

Jack W Lindsey, Conceptualization, Data curation, Software, Formal analysis, Investigation, Writing – original draft, Writing – review and editing; Elias B Issa, Conceptualization, Resources, Supervision, Funding acquisition, Methodology, Writing – original draft, Project administration, Writing – review and editing

### Author ORCIDs

Jack W Lindsey http://orcid.org/0000-0003-0930-7327
Elias B Issa http://orcid.org/0000-0002-5387-7207

Reviewer #2 (Public Review): https://doi.org/10.7554/eLife.91685.3.sa1
Reviewer #3 (Public Review): https://doi.org/10.7554/eLife.91685.3.sa2
Author response https://doi.org/10.7554/eLife.91685.3.sa3

## Additional files

### Supplementary files

• Supplementary file 1. Tables of datasets and models used. (a) Table of datasets used for measuring similarity of models to the brain. Datasets from both macaque and human high-level visual cortex as well as high-level visual behavior were collated for testing the brainlikeness of computational models. For neural and fMRI datasets, the features in the model were used to predict the image-by-image response pattern of each neuron or voxel. For behavior datasets, the performance of linear decoders built atop model representations were compared to performance per image of macaques and humans. (b) Table of models tested. For each model, we measured representational factorization and invariance in each of the final five layers of the model as well as evaluating their brainlikeness using the datasets in (a).

• MDAR checklist

### Data availability

The current manuscript is a computational study, so no data have been generated for this manuscript. Publicly available datasets and models were used. Analysis code is available at https://github.com/issalab/Lindsey-Issa-Factorization, (copy archived at *Issa, 2024*).

The following previously published datasets were used:

| Author(s) | Year | Dataset title | Dataset URL | Database and Identifier |
|---|---|---|---|---|
| Kay KN, Naselaris T, Gallant J | 2011 | fMRI of human visual areas in response to natural images | https://doi.org/10.6080/K0QN64NG | Collaborative Research in Computational Neuroscience, 10.6080/K0QN64NG |
| Shen G, Horikawa T, Majima K, Kamitani Y | 2020 | Deep Image Reconstruction | https://doi.org/10.18112/openneuro.ds001506.v1.3.1 | OpenNeuro, 10.18112/openneuro.ds001506.v1.3.1 |

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
