## [Editor Report · eLife assessment]

The study makes a **valuable** empirical contribution to our understanding of visual processing in primates and deep neural networks, with a specific focus on the concept of factorization. The analyses provide **convincing** evidence that high factorization scores are correlated with neural predictivity. This work will be of interest to systems neuroscientists studying vision and could inspire further research that ultimately may lead to better models of or a better understanding of the brain.

---

## [Referee Report · Reviewer #2 (Public Review)]

Summary:

The dominant paradigm in the past decade for modeling the ventral visual stream's response to images has been to train deep neural networks on object classification tasks and regress neural responses from units of these networks. While object classification performance is correlated to variance explained in the neural data, this approach has recently hit a plateau of variance explained, beyond which increases in classification performance do not yield improvements in neural predictivity. This suggests that classification performance may not be a sufficient objective for building better models of the ventral stream. Lindsey & Issa study the role of factorization in predicting neural responses to images, where factorization is the degree to which variables such as object pose and lighting are represented independently in orthogonal subspaces. They propose factorization as a candidate objective for breaking through the plateau suffered by models trained only on object classification. They show the degree of factorization in a model captures aspects of neural variance that classification accuracy alone does not capture, hence factorization may be an objective that could lead to better models of ventral stream. I think the most important figure for a reader to see is Fig. 6.

Strengths:

This paper challenges the dominant approach to modeling neural responses in the ventral stream, which itself is valuable for diversifying the space of ideas.

This paper uses a wide variety of datasets, spanning multiple brain areas and species. The results are consistent across the datasets, which is a great sign of robustness.

The paper uses a large set of models from many prior works. This is impressively thorough and rigorous.

The authors are very transparent, particularly in the supplementary material, showing results on all datasets. This is excellent practice.

Weaknesses:

The authors have addressed many of the weaknesses in the original review. The weaknesses that remain are limitations of the work that cannot be easily addressed. In addition to the limitations stated at the end of the discussion, I'll add two:

(1) This work shows that factorization is correlated with neural similarity, and notably explains some variance in neural similarity that classification accuracy does not explain. This suggests that factorization could be used as an objective (along with classification accuracy) to build better models of the brain. However, this paper does not do that - using factorization to build better models of the brain is left to future work.

---

## [Referee Report · Reviewer #3 (Public Review)]

Summary:

Object classification serves as a vital normative principle in both the study of the primate ventral visual stream and deep learning. Different models exhibit varying classification performances and organize information differently. Consequently, a thriving research area in computational neuroscience involves identifying meaningful properties of neural representations that act as bridges connecting performance and neural implementation. In the work of Lindsey and Issa, the concept of factorization is explored, which has strong connections with emerging concepts like disentanglement [1,2,3] and abstraction [4,5]. Their primary contributions encompass two facets: (1) The proposition of a straightforward method for quantifying the degree of factorization in visual representations. (2) A comprehensive examination of this quantification through correlation analysis across deep learning models.

To elaborate, their methodology, inspired by prior studies [6], employs visual inputs featuring a foreground object superimposed onto natural backgrounds. Four types of scene variables, such as object pose, are manipulated to induce variations. To assess the level of factorization within a model, they systematically alter one of the scene variables of interest and estimate the proportion of encoding variances attributable to the parameter under consideration.

The central assertion of this research is that factorization represents a normative principle governing biological visual representation. The authors substantiate this claim by demonstrating an increase in factorization from macaque V4 to IT, supported by evidence from correlated analyses revealing a positive correlation between factorization and decoding performance. Furthermore, they advocate for the inclusion of factorization as part of the objective function for training artificial neural networks. To validate this proposal, the authors systematically conduct correlation analyses across a wide spectrum of deep neural networks and datasets sourced from human and monkey subjects. Specifically, their findings indicate that the degree of factorization in a deep model positively correlates with its predictability concerning neural data (i.e., goodness of fit).

Strengths:

The primary strength of this paper is the authors' efforts in systematically conducting analysis across different organisms and recording methods. Also, the definition of factorization is simple and intuitive to understand.

Weaknesses:

Comments on revised version:

I thank the authors for addressing the weaknesses I brought up regarding the manuscript.

---

## [Author Response]

The following is the authors’ response to the original reviews.

**eLife assessment**
The study makes a valuable empirical contribution to our understanding of visual processing in primates and deep neural networks, with a specific focus on the concept of factorization. The analyses provide solid evidence that high factorization scores are correlated with neural predictivity, yet more evidence would be needed to show that neural responses show factorization. Consequently, while several aspects require further clarification, in its current form this work is interesting to systems neuroscientists studying vision and could inspire further research that ultimately may lead to better models of or a better understanding of the brain.
**Public Reviews:**

**Reviewer #1 (Public Review):**
Summary:The paper investigates visual processing in primates and deep neural networks (DNNs), focusing on factorization in the encoding of scene parameters. It challenges the conventional view that object classification is the primary function of the ventral visual stream, suggesting instead that the visual system employs a nuanced strategy involving both factorization and invariance. The study also presents empirical findings suggesting a correlation between high factorization scores and good neural predictivity.Strengths:(1) Novel Perspective: The paper introduces a fresh viewpoint on visual processing by emphasizing the factorization of non-class information.(2) Methodology: The use of diverse datasets from primates and humans, alongside various computational models, strengthens the validity of the findings.(3) Detailed Analysis: The paper suggests metrics for factorization and invariance, contributing to a future understanding & measurements of these concepts.Weaknesses:(1) Vagueness (Perceptual or Neural Invariance?): The paper uses the term 'invariance', typically referring to perceptual stability despite stimulus variability [1], as the complete discarding of nuisance information in neural activity. This oversimplification overlooks the nuanced distinction between perceptual invariance (e.g., invariant object recognition) and neural invariance (e.g., no change in neural activity). It seems that by 'invariance' the authors mean 'neural' invariance (rather than 'perceptual' invariance) in this paper, which is vague. The paper could benefit from changing what is called 'invariance' in the paper to 'neural invariance' and distinguish it from 'perceptual invariance,' to avoid potential confusion for future readers. The assignment of 'compact' representation to 'invariance' in Figure 1A is misleading (although it can be addressed by the clarification on the term invariance). [1] DiCarlo JJ, Cox DD. Untangling invariant object recognition. Trends in cognitive sciences. 2007 Aug 1;11(8):333-41.

Thanks for pointing out this ambiguity. In our Introduction we now explicitly clarify that we use “invariance” to refer to neural, rather than perceptual invariance, and we point out that both factorization and (neural) invariance may be useful for obtaining behavioral/perceptual invariance.

(2) Details on Metrics: The paper's explanation of factorization as encoding variance independently or uncorrelatedly needs more justification and elaboration. The definition of 'factorization' in Figure 1B seems to be potentially misleading, as the metric for factorization in the paper seems to be defined regardless of class information (can be defined within a single class). Does the factorization metric as defined in the paper (orthogonality of different sources of variation) warrant that responses for different object classes are aligned/parallel like in 1B (middle)? More clarification around this point could make the paper much richer and more interesting.

Our factorization metric measures the degree to which two sets of scene variables are factorized from one another. In the example of Fig. 1B, we apply this definition to the case of factorization of class vs. non-class information. Elsewhere in the paper we measure factorization of several other quantities unrelated to class, specifically camera viewpoint, lighting conditions, background content, and object pose. In our revised manuscript we have clarified the exposition surrounding Fig. 1B to make it clear that factorization, as we define it, can be applied to other quantities as well and that responses do not need to be aligned/parallel but simply live in a different set of dimensions whether linearly or nonlinearly arranged. Thanks for raising the need to clarify this point.

(3) Factorization vs. Invariance: Is it fair to present invariance vs. factorization as mutually exclusive options in representational hypothesis space? Perhaps a more fair comparison would be factorization vs. object recognition, as it is possible to have different levels of neural variability (or neural invariance) underlying both factorization and object recognition tasks.

We do not mean to imply that factorization and invariance are mutually exclusive, or that they fully characterize the space of possible representations. However, they are qualitatively distinct strategies for achieving behavioral capabilities like object recognition. In the revised manuscript we also include a comparison to object classification performance (Figures 5C & S4, black x’s) as a predictor of brain-like representations, alongside the results for factorization and invariance.

In our revised Introduction and beginning of the Results section, we make it more clear that factorization and invariance are not mutually exclusive – indeed, our results show that both factorization and invariance for some scene variables like lighting and background identity are signatures of brain-like representations. Our study focuses on factorization because we believe its importance has not been studied or highlighted to the degree that invariance to “nuisance” parameters has in concert with selectivity to object identity in individual neuron tuning functions. Moreover, the loss functions used for supervised training functions of neural networks for image classification would seem to encourage invariance as a representational strategy. Thus, the finding that factorization of scene parameters is an equally good if not better predictor of brain-like representations may motivate new objective functions for neural network training.

(4) Potential Confounding Factors in Empirical Findings: The correlation observed in Figure 3 between factorization and neural predictivity might be influenced by data dimensionality, rather than factorization per se [2]. Incorporating discussions around this recent finding could strengthen the paper.[2] Elmoznino E, Bonner MF. High-performing neural network models of the visual cortex benefit from high latent dimensionality. bioRxiv. 2022 Jul 13:2022-07.

We thank the Reviewer for pointing out this important, potential confound and the need for a direct quantification. We have now included an analysis computing how well dimensionality (measured using the participation ratio metric for natural images, as was done in [2] Elmoznino& Bonner bioRxiv. 2022) can account for model goodness-of-fit (additional pink bars in Figure 6). Factorization of scene parameters appears to add more predictive power than dimensionality on average (Figure 6, light shaded bars), and critically, factorization+classification jointly predict goodness-of-fit significantly better than dimensionality+classification for V4 and IT/HVC brain areas (Figure 6, dark shaded bars). Indeed, dimensionality+classification is only slightly more predictive than classification alone for V4 and IT/HVC indicating some redundancy in those measures with respect to neural predictivity of models (Figure 6, compare dark shaded pink bar to dashed line).

That said, high-dimensional representations can, in principle, better support factorization, and thus we do not regard these two representational strategies necessarily in competition. Rather, our results suggest (consistent with [2]) that dimensionality is predictive of brain-like representation to some degree, such that some (but not all) of factorization’s predictive power may indeed owe to a partial correlation with dimensionality. We elaborate in the Discussion where this point comes up and now refer to the updated Figure 6 that shows the control for dimensionality.

Conclusion:The paper offers insightful empirical research with useful implications for understanding visual processing in primates and DNNs. The paper would benefit from a more nuanced discussion of perceptual and neural invariance, as well as a deeper discussion of the coexistence of factorization, recognition, and invariance in neural representation geometry. Additionally, addressing the potential confounding factors in the empirical findings on the correlation between factorization and neural predictivity would strengthen the paper's conclusions.

Taken together, we hope that the changes described above address the distinction between neural and perceptual invariance, provide a more balanced understanding of the contributions of factorization, invariance, and local representational geometry, and rule against dimensionality for natural images as contributing to the main finding of the benefits from factorization of scene parameters.

**Reviewer #2 (Public Review):**
Summary:The dominant paradigm in the past decade for modeling the ventral visual stream's response to images has been to train deep neural networks on object classification tasks and regress neural responses from units of these networks. While object classification performance is correlated to the variance explained in the neural data, this approach has recently hit a plateau of variance explained, beyond which increases in classification performance do not yield improvements in neural predictivity. This suggests that classification performance may not be a sufficient objective for building better models of the ventral stream. Lindsey & Issa study the role of factorization in predicting neural responses to images, where factorization is the degree to which variables such as object pose and lighting are represented independently in orthogonal subspaces. They propose factorization as a candidate objective for breaking through the plateau suffered by models trained only on object classification.They claim that (i) maintaining these non-class variables in a factorized manner yields better neural predictivity than ignoring non-class information entirely, and (ii) factorization may be a representational strategy used by the brain.The first of these claims is supported by their data. The second claim does not seem well-supported, and the usefulness of their observations is not entirely clear.Strengths:This paper challenges the dominant approach to modeling neural responses in the ventral stream, which itself is valuable for diversifying the space of ideas.This paper uses a wide variety of datasets, spanning multiple brain areas and species. The results are consistent across the datasets, which is a great sign of robustness.The paper uses a large set of models from many prior works. This is impressively thorough and rigorous.The authors are very transparent, particularly in the supplementary material, showing results on all datasets. This is excellent practice.Weaknesses:(1) The primary weakness of this paper is a lack of clarity about what exactly is the contribution. I see two main interpretations: (1-A) As introducing a heuristic for predicting neural responses that improve over-classification accuracy, and (1-B) as a model of the brain's representational strategy. These two interpretations are distinct goals, each of which is valuable. However, I don't think the paper in its current form supports either of them very well:(1-A) Heuristic for neural predictivity. The claim here is that by optimizing for factorization, we could improve models' neural predictivity to break through the current predictivity plateau. To frame the paper in this way, the key contribution should be a new heuristic that correlates with neural predictivity better than classification accuracy. The paper currently does not do this. The main piece of evidence that factorization may yield a more useful heuristic than classification accuracy alone comes from Figure 5. However, in Figure 5 it seems that factorization along some factors is more useful than others, and different linear combinations of factorization and classification may be best for different data. There is no single heuristic presented and defended. If the authors want to frame this paper as a new heuristic for neural predictivity, I recommend the authors present and defend a specific heuristic that others can use, e.g. [K * factorization_of_pose + classification] for some constant K, and show that (i) this correlates with neural predictivity better than classification alone, and (ii) this can be used to build models with higher neural predictivity. For (ii), they could fine-tune a state-of-the-art model to improve this heuristic and show that doing so achieves a new state-of-the-art neural predictivity. That would be convincing evidence that their contribution is useful.

Our paper does not make any strong claim regarding the Reviewer’s point 1-A (on heuristics for neural predictivity). In the Discussion, last paragraph, we better specify that our work is merely suggestive of claim 1-A about heuristics for more neurally predictive, more brainlike models. We believe that our paper supports the Reviewer’s point 1-B (on brain representation) as we discuss below.

We leave it to future work to determine if factorization could help optimize models to be more brainlike. This treatment may require exploration of novel model architectures and loss functions, and potentially also more thorough neural datasets that systematically vary many different forms of visual information for validating any new models.

(1-B) Model of representation in the brain. The claim here is that factorization is a general principle of representation in the brain. However, neural predictivity is not a suitable metric for this, because (i) neural predictivity allows arbitrary linear decoders, hence is invariant to the orthogonality requirement of factorization, and (ii) neural predictivity does not match the network representation to the brain representation. A better metric is representational dissimilarity matrices. However, the RDM results in Figure S4 actually seem to show that factorization does not do a very good job of predicting neural similarity (though the comparison to classification accuracy is not shown), which suggests that factorization may not be a general principle of the brain. If the authors want to frame the paper in terms of discovering a general principle of the brain, I suggest they use a metric (or suite of metrics) of brain similarity that is sensitive to the desiderata of factorization, e.g. doesn't apply arbitrary linear transformations, and compare to classification accuracy in addition to invariance.

We agree with the Reviewer about the shortcomings of neural predictivity for comparing representational geometries, and in our revised manuscript we have provided a more comprehensive set of results that includes RDM predictivity in new Figures 6 & 7, alongside the results for neural fit predictivity. In addition, as suggested we added classification accuracy predictivity in Figures 5C & S4 (black x’s) for visual comparison to factorization/invariance. In Figure S4 on RDMs, it is apparent how factorization is at least as good a predictor as classification on all V4 & IT datasets from both monkeys and humans (compared x’s to filled circles in Figure S4; note that some of the points from the original Figure S4 changed as we discovered a bug in the code that specifically affected the RDM analysis for a few of the datasets).

We find that the newly included RDM analyses in Figures 6 & 7 are consistent with the conclusions of the neural fit regression analyses: that the correlation of factorization metrics with RDM matches are strong, comparable in magnitude to that of classification accuracy (Figure 6, 3rd & 4th columns, compare black dashed line to faded colored bars) and are not fully accounted for by the model’s classification accuracy alone (Figure 6, 3rd & 4th columns, higher unfaded bars for classification combined with factorization, and see corresponding example scatters in Figure 7 middle/bottom rows).

It is encouraging that the added benefit of factorization for RDM predictivity accounting for classification performance is at least as good as the improvement seen for neural fit predictivity (Figure 6, 1st & 2nd columns for encoding fits versus 3rd & 4th columns for RDM correlations).

(2) I think the comparison to invariance, which is pervasive throughout the paper, is not very informative. First, it is not surprising that invariance is more weakly correlated with neural predictivity than factorization, because invariant representations lose information compared to factorized representations. Second, there has long been extensive evidence that responses throughout the ventral stream are not invariant to the factors the authors consider, so we already knew that invariance is not a good characterization of ventral stream data.

While we appreciate the Reviewer’s intuition that highly invariant representations are not strongly supported in the high-level visual cortex, we nevertheless thought it was valuable to put this intuition to a quantitative, detailed test. As a result, we uncovered effects that were not obvious a priori, at least to us – for example, that invariance for some scene parameters (camera view, object pose) is negatively correlated with neural predictions while invariance to others (background, lighting) is positively correlated. Thus, our work exercises the details of invariance for different types of information.

(3) The formalization of the factorization metric is not particularly elegant, because it relies on computing top K principal components for the other-parameter space, where K is arbitrarily chosen as 10. While the authors do show that in their datasets the results are not very sensitive to K (Figure S5), that is not guaranteed to be the case in general. I suggest the authors try to come up with a formalization that doesn't have arbitrary constants. For example, one possibility that comes to mind is E[delta_a x delta_b], where 'x' is the normalized cross product, delta_a, and delta_b are deltas in representation space induced by perturbations of factors a and b, and the expectation is taken over all base points and deltas. This is just the first thing that comes to mind, and I'm sure the authors can come up with something better. The literature on disentangling metrics in machine learning may be useful for ideas on measuring factorization.

Thanks to the Reviewer for raising this point. First, we wish to clarify a potential misunderstanding of the factorization metric: the number K of principal components we choose is not an arbitrary constant, but rather calibrated to capture a certain fraction of variance, set to 90% by default in our analyses. While this variance threshold is indeed an arbitrary hyperparameter, it has a more intuitive interpretation than the number of principal components.

Nonetheless, the Reviewer’s comment did inspire us to consider another metric for factorization that does not depend on any arbitrary parameters. In the revised version, we now include a covariance matrix based metric which simply measures the elementwise correlation of the covariance matrices induced by varying the scene parameter of interest and the covariance matrix induced by varying the other parameters (and then subtracts this quantity from 1).

Correspondingly, we now present results for both the new covariance based measure and the original PCA based one in Figures 5C, 6, and 7. The main findings remain largely the same when using the covariance based metric, and the covariance based metric (Figure 5C, compare light shaded to dark shaded filled circles; Figure 6, compare top row to bottom row; Figure 7, compare middle rows to bottom rows).

Ultimately, we believe these two metrics are complementary and somewhat analogous to two metrics commonly used for measuring dimensionality (the number of components needed to explain a certain fraction of the variance, analogous to our original PCA based definition; the participation ratio, analogous to our covariance based definition). We have added the formula for the covariance based factorization metric along with a brief description to the Methods.

(4) The authors defined the term "factorization" according to their metric. I think introducing this new term is not necessary and can be confusing because the term "factorization" is vague and used by different researchers in different ways. Perhaps a better term is "orthogonality", because that is clear and seems to be what the authors' metric is measuring.

We agree with the Reviewer that factorization has become an overloaded term. At the same time, we think that in this context, the connotation of the term factorization effectively conveys the notion of separating out different latent sources of variance (factors) such that they can be encoded in orthogonal subspaces.

To aid clarity, we now mention in the Introduction that factorization defined here is meant to measure orthogonalization of scene factors. Additionally, in the Discussion section, we now go into more detail comparing our metric to others previously used in the literature, including orthogonality, to help put it in context.

(5) One general weakness of the factorization paradigm is the reliance on a choice of factors. This is a subjective choice and becomes an issue as you scale to more complex images where the choice of factors is not obvious. While this choice of factors cannot be avoided, I suggest the authors add two things: First, an analysis of how sensitive the results are to the choice of factors (e.g. transform the basis set of factors and re-run the metric); second, include some discussion about how factors may be chosen in general (e.g. based on temporal statistics of the world, independent components analysis, or something else).

The Reviewer raises a very reasonable point about the limitation of this work. While we limited our analysis to generative scene factors that we know about and that could be manipulated, there are many potential factors to consider. It is not clear to us exactly how to implement the Reviewer’s suggestion of transforming the basis set of factors, as the factors we consider are highly nonlinear in the input space. Ultimately, we believe that finding unsupervised methods to characterize the “true” set of factors that is most useful for understanding visual representations is an important subject for future work, but outside the scope of this particular study. We have added a comment to this effect in the Discussion.

**Reviewer #3 (Public Review):**
Summary:Object classification serves as a vital normative principle in both the study of the primate ventral visual stream and deep learning. Different models exhibit varying classification performances and organize information differently. Consequently, a thriving research area in computational neuroscience involves identifying meaningful properties of neural representations that act as bridges connecting performance and neural implementation. In the work of Lindsey and Issa, the concept of factorization is explored, which has strong connections with emerging concepts like disentanglement [1,2,3] and abstraction [4,5]. Their primary contributions encompass two facets: (1) The proposition of a straightforward method for quantifying the degree of factorization in visual representations. (2) A comprehensive examination of this quantification through correlation analysis across deep learning models.To elaborate, their methodology, inspired by prior studies [6], employs visual inputs featuring a foreground object superimposed onto natural backgrounds. Four types of scene variables, such as object pose, are manipulated to induce variations. To assess the level of factorization within a model, they systematically alter one of the scene variables of interest and estimate the proportion of encoding variances attributable to the parameter under consideration.The central assertion of this research is that factorization represents a normative principle governing biological visual representation. The authors substantiate this claim by demonstrating an increase in factorization from macaque V4 to IT, supported by evidence from correlated analyses revealing a positive correlation between factorization and decoding performance. Furthermore, they advocate for the inclusion of factorization as part of the objective function for training artificial neural networks. To validate this proposal, the authors systematically conduct correlation analyses across a wide spectrum of deep neural networks and datasets sourced from human and monkey subjects. Specifically, their findings indicate that the degree of factorization in a deep model positively correlates with its predictability concerning neural data (i.e., goodness of fit).Strengths:The primary strength of this paper is the authors' efforts in systematically conducting analysis across different organisms and recording methods. Also, the definition of factorization is simple and intuitive to understand.Weaknesses:This work exhibits two primary weaknesses that warrant attention: (i) the definition of factorization and its comparison to previous, relevant definitions, and (ii) the chosen analysis method.Firstly, the definition of factorization presented in this paper is founded upon the variances of representations under different stimuli variations. However, this definition can be seen as a structural assumption rather than capturing the effective geometric properties pertinent to computation. More precisely, the definition here is primarily statistical in nature, whereas previous methodologies incorporate computational aspects such as deviation from ideal regressors [1], symmetry transformations [3], generalization [5], among others. It would greatly enhance the paper's depth and clarity if the authors devoted a section to comparing their approach with previous methodologies [1,2,3,4,5], elucidating any novel insights and advantages stemming from this new definition.[1] Eastwood, Cian, and Christopher KI Williams. "A framework for the quantitative evaluation of disentangled representations." International conference on learning representations. 2018.[2] Kim, Hyunjik, and Andriy Mnih. "Disentangling by factorising." International Conference on Machine Learning. PMLR, 2018.[3] Higgins, Irina, et al. "Towards a definition of disentangled representations." arXiv preprint arXiv:1812.02230 (2018).[4] Bernardi, Silvia, et al. "The geometry of abstraction in the hippocampus and prefrontal cortex." Cell 183.4 (2020): 954-967.[5] Johnston, W. Jeffrey, and Stefano Fusi. "Abstract representations emerge naturally in neural networks trained to perform multiple tasks." Nature Communications 14.1 (2023): 1040.

Thanks to the Reviewer for this suggestion. We agree that our initial submission did not sufficiently contextualize our definition of factorization with respect to other related notions in the literature. We have added additional discussion of these points to the Discussion section in the revised manuscript and have included therein the citations provided by the Reviewer (please see the third paragraph of Discussion).

Secondly, in order to establish a meaningful connection between factorization and computation, the authors rely on a straightforward synthetic model (Figure 1c) and employ multiple correlation analyses to investigate relationships between the degree of factorization, decoding performance, and goodness of fit. Nevertheless, the results derived from the synthetic model are limited to the low training-sample regime. It remains unclear whether the biological datasets under consideration fall within this low training-sample regime or not.

We agree that our model in Figure 1C is very simple and does not fully capture the complex interactions between task performance and features of representational geometry, like factorization. We intend it only as a proof of concept to illustrate how factorized representations can be beneficial for some downstream task use cases. While the benefits of factorized representations disappear for large numbers of samples in this simulation, we believe this is primarily a consequence of the simplicity and low dimensionality of the simulation. Real-world visual information is complex and high-dimensional, and as such the relevant sample size regime in which factorization offers tasks benefits may be much greater. As a first step toward this real-world setting, Figure 2 shows how decreasing the amount of factorization in neural population data in macaque V4/IT can have an effect on object identity decoding.

**Recommendations for the authors**

**Reviewer #1 (Recommendations For The Authors):**
Missing citations: The paper could benefit from discussions & references to related papers, such as:Higgins I, Chang L, Langston V, Hassabis D, Summerfield C, Tsao D, Botvinick M. Unsupervised deep learning identifies semantic disentanglement in single inferotemporal face patch neurons. Nature communications. 2021 Nov 9;12(1):6456.

We have added additional discussion of related work, including the suggested reference and others on disentanglement, to the Discussion section in the revised manuscript.

**Reviewer #2 (Recommendations For The Authors):**
Here are several small recommendations for the authors, all much more minor than those in the public review:I suggest more use of equations in methods sections about Figure 1C and macaque neural data analysis.

Thanks for this suggestion. We have added new Equation 1 for the method transforming neural data to reduce factorization of a variable while preserving other firing rate statistics.

In Figure 1-C, the methods indicate that Gaussian noise was added. This is a very important detail, and complexifies the interpretation of the figure because it adds an assumption about the structure of noise. In other words, if I understand correctly, the correct interpretation of Figure 1C is "assuming i.i.d. noise, decoding accuracy improves with factorization." The i.i.d. noise is a big assumption, and it is debated how well the brain satisfies this assumption. I suggest you either omit noise for this figure or clearly state in the main text (e.g. caption) that the figure must be interpreted under an i.i.d. noise assumption.

We have added an explicit statement of the i.i.d. noise assumption to the Figure 1C legend.

For Figure 2B, I suggest labeling the x-axis clearly below the axis on both panels. Currently, it is difficult to read, particularly in print.

We have made the x-axis labels more clear and included on both panels.

Figure 3A is difficult to read because of the very small task. I suggest avoiding such small fonts.

We agree that Figure 3A is difficult to read. We have broken out Figure 3 into two new Figures 3 & 4 to increase clarity and sizing of text in Figure 3A.

**Reviewer #3 (Recommendations For The Authors):**
To strengthen this work, it is advisable to incorporate more comprehensive comparisons with previous research, particularly within the machine learning (ML) community. For instance, it would be beneficial to explore and reference works focusing on disentanglement [1,2,3]. This would provide valuable context and facilitate a more robust understanding of the contributions and novel insights presented in the current study.

We have added additional discussion of related work and other notions similar to factorization to the Discussion section in the revised manuscript.

Additionally, improving the quality of the figures is crucial to enhance the clarity of the findings:Figure 2: The caption of subfigure B could be revised for greater clarity.

Thank you, we have substantially clarified this figure caption.

Figure 3: Consider a more equitable approach for computing the correlation coefficient, such as calculating it separately for different types of models. In the case of supervised models, it appears that the correlation between invariance and goodness of fit may not be negligible across various scene parameters.

We appreciate the suggestion, but we are not confident in our ability to conclude much from analyses restricted to particular model classes, given the relatively small N and the fact that the different model classes themselves are an important source of variance in our data.

Figure 4: To enhance the interpretability of subfigures A and B, it may be beneficial to include p-values (indicating confidence levels).

As we supply bootstrapped confidence intervals for our results, which provide at least as much information as p-values, and most of the effects of interest are fairly stark when comparing invariance to factorization, p-values were not needed to support our points. We added a sentence to the legend of new Figure 5 (previously Figure 4) indicating that error bars reflect standard deviations over bootstrap resampling of the models.

Figure 5: For subfigure B, it could be advantageous to plot the results solely for factorization, allowing for a clear assessment of whether the high correlation observed in Classification+Factorization arises from the combined effects of both factors or predominantly from factorization alone.

First, we clarify/note that the scatters solely for factorization that the Reviewer seeks are already presented earlier in the manuscript across all conditions in Figures 4A,B and Figure S2.

While we could also include these in new Figure 7 (previously Figure 5B) as the Reviewer suggests, we believe it would distract from the message of that figure at the end of the manuscript – which is that factorization is useful as a supplement to classification in predictive matches to neural data. Nonetheless, new Figure 6 (old Figure 5A) provides a summary quantification of the information that the reviewer requests (Fig. 6, faded colored bars reflect the contribution of factorization alone).